# Comparison of Slate Safety Wearable Device to Ingestible Pill and Wearable Heart Rate Monitor

**DOI:** 10.3390/s23020877

**Published:** 2023-01-12

**Authors:** Michael Callihan, Heather Cole, Holly Stokley, Joshua Gunter, Kaitlyn Clamp, Alexis Martin, Hannah Doherty

**Affiliations:** Capstone College of Nursing, University of Alabama, Tuscaloosa, AL 35487, USA

**Keywords:** wearable device, heat stress, occupational health

## Abstract

Background: With the increase in concern for deaths and illness related to the increase in temperature globally, there is a growing need for real-time monitoring of workers for heat stress indicators. The purpose of this study was to determine the usability of the Slate Safety (SS) wearable physiological monitoring system. Methods: Twenty nurses performed a common task in a moderate or hot environment while wearing the SS device, the Polar 10 monitor, and having taken the e-Celsius ingestible pill. Data from each device was compared for correlation and accuracy. Results: High correlation was determined between the SS wearable device and the Polar 10 system (0.926) and the ingestible pill (0.595). The SS was comfortable to wear and easily monitored multiple participants from a distance. Conclusions: The Slate Safety wearable device demonstrated accuracy in measuring core temperature and heart rate while not restricting the motion of the worker, and provided a remote monitoring platform for physiological parameters.

## 1. Introduction

According to the National Heart, Lung, and Blood Institute (NHLBI), an average of 1373 extra deaths per year in the United States can be attributed to weather with a heat index at or above 90 degrees [1]. While this extreme temperature is becoming more prevalent in the world, it has been common within many work environments for a long time. The US Department of Labor reports that between 2011 and 2019 there were 344 work-related deaths due to environmental exposure [2]; however, this number is suspected to be much less representative of the actual number of workplace injuries and deaths related to heat illness [3]. The physical activity of a worker increases the metabolic energy turnover, thus increasing their body temperatures [4], making them more susceptible to heat-related stress on the body.

Heat stress occurs when the body cannot get rid of excess heat, placing a worker at an increased risk for illness or injury [3]. Common symptoms of heat exhaustion include headache, nausea, weakness, dizziness, elevated body temperatures, and decreased urine output [3]. Heat stress has also been shown to increase systemic inflammation within the body, which can be linked to multiple common health problems [5,6]. With the negative immediate and delayed impacts of this systemic inflammatory response, heat stress represents a major concern for many workers’ chronic disease processes. Given the impact of increased temperatures within the various climates worldwide, we must have the means to monitor the effects of heat on workers [7]. 

According to the National Safety Council, 49% of employers have implemented some sort of wearable technology to track the health risks of their workers [8] and reportedly monitor skin temperature and sweat rates. The body’s natural response to a heat-stress environment is to increase the blood circulating to the surface of the skin to promote cooling through the sweating process. To accomplish this, the heart rate will increase and blood vessels will dilate, resulting in a decrease in blood pressure [9,10]. This dangerous fluctuation in important vital signs that are representative of physiological functioning makes it clear that employers need to implement a wearable device that can monitor both body temperature and heart rate simultaneously to provide early detection of the effects of heat stress on the body. Early detection could be the key to lowering the number of work-related deaths that are caused by working in excessive environmental conditions such as heat.

Current practices to limit an employee’s exposure to heat-related stressors revolve around the environmental factors of temperature and humidity, combined with clothing worn and the level of exertion, to estimate the levels of heat stress placed on the worker [11]. Organizations such as the American Conference of Governmental and Industrial Hygiene (ACGIH) set threshold limit values (TLV) based on these variables, but the response of the worker to heat is not individualized to their specific situations [12]. While heat stress prevention based on the environmental condition is a start in mitigating heat injury, the monitoring of each workers’ physiological parameters is critical in truly preventing heat related injury [12]. 

Wearable sensor systems require three main components in order to remotely monitor an individual: (1) the data collection hardware, (2) the communication hardware, and (3) the analysis techniques to interpret the data [13]. Wearable devices related to heat stress prevention includes sensors that monitor the sweat production [14,15,16], temperature (core, skin, rectal) [12,17], heart rate, variability, recovery rate [12], and perceived exertion [12]. Multiple physiological parameters can be collected utilizing smart wearable sensors to measure electrocardiograph (ECG) tracing, blood pressure, pulse pressure, respiratory rate, and pulse oximetry [18,19,20,21,22,23] which is recorded in a data acquisition device and transmitted to a monitoring location.

Body temperature sensing technologies frequently utilize either photoconductivity, infrared, thermistor, thermoelectric effects, or photodetector technologies to collect the raw data [24,25]. Utilizing these technologies, the estimation of the core temperature based on the skin temperature and perfusion to the skin was shown to be highly accurate (MAE 0.297, MSE 0.133) [26]. Core temperature can also be monitored using a radiometer to measure the thermal radiation emitted from the body with high levels of accuracy [27]. 

With limited devices available to remotely monitor combined heart rate, exertion, and body temperature, this pilot study aims to determine the usability of the Slate Safety physiological monitor (SS) wearable sensor based on accuracy, comfort, and the ability to continuously monitor a worker. This was determined by measuring the correlation between the SS wearable sensor system and the ingestible core temperature monitoring device, evaluating the comfort of the device with each participant, and through the continuous monitoring of the participant by the research team.

## 2. Materials and Methods

Study design. Following IRB approval, participants were recruited via social media and word of mouth until a total of twenty was achieved for the pilot study. This pilot study was a randomized control trial conducted while nurses performed cardiopulmonary resuscitation (CPR) on a simulation mannequin in either a hot environment or a moderate environment. The number of participants for this pilot study was limited based on the availability of equipment needed for data collection. Inclusion criteria for the study were being a licensed nurse or in the final semester of a four-year nursing program, between the ages of 18 and 60 years, having no history of heat-related illness, and being capable of performing CPR for fifteen minutes on a simulated patient. Participants met with the research team the night before collections to be screened for the study, complete a urine pregnancy test, and receive their ingestible pill. All participants ingested the pill 4–6 h before reporting to the simulation laboratory for collection. Upon arrival to the simulation center, informed consent was obtained, and participants were fitted with monitoring devices. The e-Celsius pills were cleared of data by having the participants wear the monitor device until the stored data read zero. 

Participants completed two cognitive tasks on a computer and two nursing skills (inserting a Foley catheter and obtaining intravenous access) before beginning the simulation. Participants then went into the simulation room with their assigned partner. The simulation began with the participants waiting for five minutes before beginning CPR on a simulation mannequin. This five-minute period was necessary to allow the participants’ bodies to acclimatize to the temperature of the room before beginning the task. Participants took turns performing CPR, rotating every two minutes, for a total of fifteen minutes. Data were collected using the SS wearable device, which collected both heart rate and temperature data, the e-Celsius ingestible pill, which collected temperature data, and the Polar 10 monitor, which collected heart rate data. Since both the e-Celsius pill and the Polar 10 heart rate devices have been shown to be accurate and are commonly used in heat stress studies, the data collected from these two systems were used as a baseline to compare the SS wearable device.

Setting. The study occurred in the College of Nursing simulation center at the investigator’s home institution. The simulation occurred in a room equipped as a hospital room with the temperature controlled to either 71 °F (moderate condition) or 85 °F (hot condition) and at a humidity of approximately 40%, depending upon the condition assigned to the group. CPR was performed on a high-fidelity simulator with the participant on a step stool and the bed at a height comfortable to the participant. Participants in the hot room also wore splash-resistant gowns for the collection to simulate the occupational conditions more accurately.

Variables. Participants were placed in pairs based on the order they were recruited and then randomly selected to complete the study in either the hot condition or the moderate condition. Randomization occurred by placing the ten group numbers into a hat and drawing out five numbers to participate in the study in the hot environment. Variables of interest for the study were the heart rate and core temperature measurements using the SS system with comparison to the Polar 10 and e-Celsius ingestible pill systems. 

Data sources/measurement. Data collected included heart rate and core temperature. Heart rate was collected using the Polar 10 heart rate monitor attached to the chest of the participant and the SS wearable device secured to the right upper arm. Core temperature was collected with the e-Celsius ingestible pill taken 4–6 h before data collection and the SS wearable device. Participants were questioned about the comfort of the SS wearable device after the collection period. At least two members of the research team performed continuous monitoring of all parameters. 

Polar 10 system. The Polar 10 system employs a chest strap to secure the device to the participant and records raw ECG and RR intervals with a resolution of 1 ms with high accuracy [28,29,30]. Variations in heart rate accuracy have been noted during different physical activities with the Polar 10 system [28,29,30]. To control for these differences, all the nursing activities for all participants were the same: the insertion of a Foley catheter, establishing IV access, and fifteen minutes of CPR on a simulation mannikin. 

Slate Safety System. The SS system utilizes a strap to attach the device to the outer aspect of the upper arm, midway between the elbow and the distal aspect of the lateral deltoid (Figure 1). The system provides for real-time monitoring of heart rate, core temperature, and exertion level in employees. The system provides remotely monitored continuous updates at fifteen second intervals of multiple members of a workgroup with a battery life of over twenty-four hours with a full charge [31]. Signals were transmitted through a cellular connection and can be stored within the unit for upload once the worker returns to an area with cellular coverage. Safety alert limits can be set for each employee independent of others being monitored. 

Bias. Comfort and ease of continuous monitoring of the participants’ physiologic data had a potential for researcher bias. To control for this, participants were asked for their thoughts on the wearable device in a private setting to determine the comfort and any restrictions on task performance. The continuous monitoring of the physiological data was conducted by at least two members of the research team and discussed with the research team following each participant. 

Study size. The sample size for the pilot study was 20 participants, with 10 participants in each group. The sample size for this pilot study was chosen based on the availability of ingestible pills and will be used to calculate the sample size for future studies. 

Quantitative variables. Variables of interest for this study included heart rate and temperature as collected using the SS wearable device compared to the e-Celsius ingestible pill and the Polar 10 heart rate monitor. Data were collected within the SS system every minute, the e-Celsius reported data at 30-s intervals, and the Polar 10 system reported data every second. All data were normalized to the minute for comparison. 

Qualitative data were collected from each participant concerning the comfort of the wearable devices as well as any restrictions on the range of motion during the completion of the tasks. Participants were asked to rate the comfort of the SS, the Polar 10 system, and taking the ingestible pill on a scale of 1–10, with ten representing the highest level of comfort. The range of motion was determined as large impairment, moderate impairment, minimal impairment, and no impairment based on normal range of motion. Members of the research team provided feedback about the ability to continuously monitor the participants in real-time during the simulation experience. 

Statistical methods. The SPSS version 27 software package was utilized for all statistical analysis. Bivariate Pearson correlations were conducted for each participant and as a total of all participants. Bland-Altman plots were completed with linear regressions performed. Accuracy was determined as a percentage of the expected value for both peaks and averages, with the ingestible pill and Polar 10 systems serving as the expected values. 

## 3. Results

### 3.1. Participants

Fifteen registered nurses, one licensed practical nurse, and four recent RN graduates not yet licensed completed the study. Most of the participants were female (85%) with an average age of 31.45 years. Table 1 provides characteristic data for all participants and the group to which they were assigned for data collection.

### 3.2. Qualitative Data

All participants reported a high level of comfort with the SS and Polar systems, with a mean score of 9.8 and 9.7, respectively. Participants reported a mean comfort level of 6.5 for taking the ingestible pill with six potential participants declining to participate in the study due to anxiety about taking the pill. These participants were educated about the safety and potential risks of the pill yet still refused participation.

All participants reported the range of motion for the SS as demonstrating no impairment from normal. Sixteen participants reported that the Polar system demonstrated no impairment to the normal range of motion, two reported minimal impairment of range-of-motion, and two participants could not wear the system due to the chest strap not being large enough. The elastic band used to secure the Polar 10 was too small to fit around the chest of two participants. 

Although both systems provided the capability for real-time monitoring of heart rate, the Polar 10 system dropped data on occasion while the participant was performing CPR. While the SS system allowed for multiple participants to be monitored through one computer, the Polar 10 system required a different device to collect the data from each system with accounts associated to each device. 

Data were collected using the e-Celsius ingestible pill, the Polar 10 heart rate monitor, and the Slate Safety wearable physiological monitor, and are reported in Table 2 below. The peak temperature with the e-Celsius ingestible pill was 38.41 °C, and with the SS wearable it was 38.43 °C. The peak heart rate collected with the Polar 10 system was 182.5 beats per minute and with the SS system it was 186 beats per minute.

Heart rate was monitored in all participants comparing data from the SS wearable to the Polar 10 device with differences and means plotted in a Bland-Altman plot (Figure 2). Linear regression demonstrated a significant bias (t = −2.6, *p* = 0.009), with the heart rate being higher for the SS than the Polar system, with a mean difference of −0.23 (meaning SS heart rate is on average 0.23 beats higher) and a standard deviation of 9.4 (95% confidence interval −18.6, 18.1). 

Correlations for the heart rate between the SS and the Polar 10 system are noted in Table 3. Data from the ingestible pill and Polar 10 system was not retained in five participants despite the systems operating during the collection of data. However, the SS System was successful in collecting data from all participants. A significant correlation of 0.926 (*p* < 0.001) with a 95% confidence interval of 0.915–0.935 was noted between the SS and Polar 10 systems during the collection period, with an intraclass correlation coefficient (ICC) of 0.961 (*p* < 0.001, 95% CI 0.955–0.966). Aggregate heart rate data demonstrated a Root Mean Square Error (RMSE) of 9.74, a Mean Absolute Error (MAE) of 5.10, and a Mean Bias Error of 0.35 beats per minute higher. Accuracy, as determined by the percentage in the difference between the expected (Polar 10) and the collected (SS), ranged from −16.7% to a 1.84% difference for the peak heart rate and −1.73% to 5.06% for the average heart rate. The accuracy for the average of all heart rates was −1.65% and 1.08% for the peak and average heart rates, respectively. 

Core temperature monitoring was completed for all participants using the SS wearable device and the e-Celsius ingestible pill, with differences and means plotted in a Bland-Altman plot (Figure 3). Linear regression demonstrated a significant bias (t = −4.6, *p* < 0.001) toward the SS, which reported a higher temperature. The mean difference in temperature between the ingestible pill and the SS was −0.03 (meaning the SS temperature is on average 0.03 C higher) with a standard deviation of 0.3 (95% confidence interval −0.62, 0.57)

Correlations between the ingestible pill and the SS for temperatures are demonstrated in Table 4. Data was not available from the ingestible pill for two participants and three participants had minimal available data. A total of 884 min of data was recorded from the SS system with at least 15 participants having more than 4 min of available data. Aggregated data demonstrated a mean error bias of 0.04 C higher for the SS system with an RMSE of 0.27 C and an MAE of 0.25 C. A significant correlation (0.595, *p* < 0.001, 95% CI 0.550–0.635; ICC 0.742, *p* < 0.001, CI 0.705–0.774) was noted between the SS wearable and the ingestible pill. Accuracy as determined by the percentage difference between the expected (e-Celsius) and the collected (SS) ranged from −1.87% to 1.29% for the peak and −1.45% to 1.98% for the average. The overall accuracy for the averages of the whole were −0.13% and 0.09% for the peak and average core temperatures, respectively. 

## 4. Discussion

The sex of the worker has shown to significantly impact the effect of temperature on the physiological response of the worker [32,33]. Participants were recruited local to the PIs academic institutions and are representative of the make-up of the nursing workforce both nationally and locally (roughly 10–16% male) [34]. While the scope of this project was to determine the usability of the Slate Safety system, differences in this response to heat will be necessary when developing specific safe work ranges based on the demonstrated core temperature, heart rate, and exertion levels of the workers. 

Acclimatization of a worker to a hot environment is critical in limiting the impact of heat stress. While it is standard to have a worker progress their workload over time to become acclimatized [35], this approach is not specific to each worker. It is known that as a person becomes acclimatized to hot environments, their cardiovascular function improves, with heart rate, heart rate recovery rate, and cardiac output all improving [35,36] which allows the body to become more efficient at ridding itself of heat [35]. Acclimatization requires physical activities to occur in an environment similar to that of the workspace, and typically takes 7–14 days to accomplish [11]. While this standard serves as a guideline, the individual response of a worker to the hot environment is not considered. 

Wearable systems have shown promise in measuring physiological parameters that can be impacted by heat stress such as heart rate, respiratory rate, cardiac function, and core temperature [10,13,18,19,20,21,22,23,24,25,26,27], which are valuable predictors of acclimatization to the hot environment. While several of these monitoring systems include multiple technologies built into a shirt or vest, the Slate Safety wearable sensor is attached via an elastic band on the upper arm, and allows for the simultaneous monitoring of estimated core temperature, heart rate, and exertion levels. The use of the band rather than a shirt material has the benefit of not adding to the layer of clothing potentially trapping heat while being reusable without the need for laundering between use. The bands are easily wiped off and ready for use during the next shift. 

This pilot study demonstrated a strong correlation and agreement in the core temperature and heart rate provided by the SS wearable device when compared to the e-Celsius ingestible pill and the Polar 10 heart rate monitoring system. Accuracy was also high for most participants for both the heart rate and temperature. While the Bland-Altman test demonstrated a bias for both temperature and heart rate readings higher with the SS, when monitoring the well-being of an employee, this was a desired finding because estimating high provides for a higher level of protection from heat stress. The results of this pilot support the idea that the SS wearable device has the necessary accuracy to be implemented into heat stress monitoring in the workplace. However, to increase the generalizability of the results, a study with a larger sample size and fewer temperature restrictions should be conducted.

Given this strong correlation and accuracy, one must next turn to examine the ability to use this device in the workplace based on the comfort of use and the ability to remotely monitor the physiology of the workers in real-time. 

### 4.1. Comfort

The comfort and range of motion while wearing the SS system performed at a high level. All participants reported that they would prefer the wearable device to the ingestible pill for core temperature monitoring. The addition of the chest strap required for the Polar 10 system did not increase discomfort for most participants; however, the chest strap utilized during the testing was not large enough to fit all participants. The chest strap utilized in the study was medium to double extra-large (67–95 cm), and was too small for two participants. In contrast, the SS system fit all participants, which suggests that it may be a better option, as it is more inclusive to all workers no matter their size.

Similar multiple physiological parameter monitoring systems have the participant wear a shift of some sort [18,19] which poses concerns in the workplace. The collection and transmission hardware must be worn by the individual, adding to the weight of the device as well as increasing the chance of the device becoming tangled or caught up on other material in the workplace. The addition of the shirt or vest also has the potential to add to the clothing layers worn by the worker, specifically in work environments that require the wearing of flame resistant personal protective equipment. The need to launder the shirt or vest between uses also poses additional difficulties for the worker, requiring them to either have multiple shirts, or to wash the shirt between each shift. The band used to attach the SS to the worker is small in nature and easily cleaned between uses. 

Another concern for a worker is any limitation on the range of motion while wearing any monitoring system. Some currently available systems utilize the wearing of a snugly fitting shirt [18,19], while others utilize a chest strap [20,21,27] or wrist band [26]. The placement of the SS midway between the elbow and the bottom of the lateral deltoid muscle allows for the full range of motion of the elbow and the shoulder while limiting the restriction of the chest or the added potential for heat being trapped close to the body by the extra layer of clothing. 

### 4.2. Continuous Monitoring

In comparison to the other two systems, the continuous monitoring ability of the SS system performed well. The SS system was set to report temperature and heart rate at 15-s intervals during the monitoring of the simulation. Monitoring the physiological parameters through the SS system allowed for monitoring from a longer range through a relay system. While the range was not a consideration for this study, it should be noted that the current version of the SS system allows for monitoring through a cloud-based service with the capability of storing data until Wi-Fi service is available. 

Temperature readings were updated periodically with the e-Celsius pills when the participant was within range of the unit for real-time monitoring. Real-time monitoring of the core temperature of participants was not available while the participant was not within a few feet of the monitoring device, making remote monitoring not possible. Data were stored on the pill until the participant placed the monitor on their body with the provided lanyard following the completion of the simulation. Data were reported in 15-s intervals when downloaded to the monitor device for analysis. The inability for real-time monitoring with these pills is concerning due to the inability to actively monitor heat stress levels allowing employers to know when an employee needs to be pulled out of a hot environment to prevent heat exhaustion.

The Polar 10 system monitored the heart rate from a distance well. Data points were dropped using the Polar system at points when the participant was actively doing compressions. This is concerning given the idea that this was the working time of the simulation, which is the highest level of risk for the participant. 

The remote monitoring of a wearable system requires a collection of hardware as well as a system capable of transferring the data to the analysis platform [13]. While some systems require external hardware be worn to collect and transmit this data [18,19,21], the SS system has this hardware built into the wearable device. This addition decreases the overall weight of the system and reduces the risk of entanglement or the snagging of an additional component in the workspace. 

### 4.3. Limitations

The limitations of this study included the sample size and the relatively low temperature the nurses were exposed to. The sample size was limited due to this being a pilot study. The temperature was restricted based on the protection of the human subjects during the study. Future studies should be conducted in work environments to determine the correlation of the systems while in the normal environment and performing routine work tasks. 

## 5. Conclusions

The SS system provided a comfortable, accurate, alternative method of monitoring the core temperature and heart rate of nurses in a simulation environment. The monitoring system was accurate in both the simulated hot and moderate environments, with no limitations based on the use of impermeable personal protective equipment. While proving superiority was not the goal of the study and cannot be stated, it is worth noting that the other systems had more limitations when it came to real-time monitoring during the times that the participants were at the highest risk for heat exhaustion. While there are limitations to the findings of this pilot study due to the small sample size and restricted temperatures, there is promise that the system will perform well in other environments.

### Generalizability

The limitations placed on this pilot study do not allow for the generalization of the findings to all populations. Given the current accuracy and usability of the system in a controlled environment, future studies should be run in a natural setting to compare the performance of the SS system to the accuracy of existing systems under normal working conditions. 

## Figures and Tables

**Figure 1 sensors-23-00877-f001:**
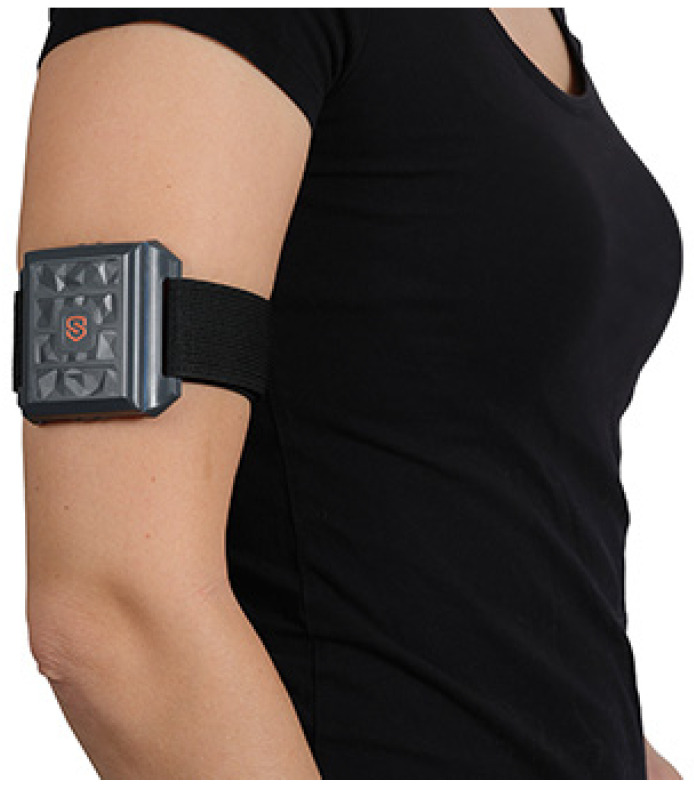
Picture of Slate Safety sensor on participant [31].

**Figure 2 sensors-23-00877-f002:**
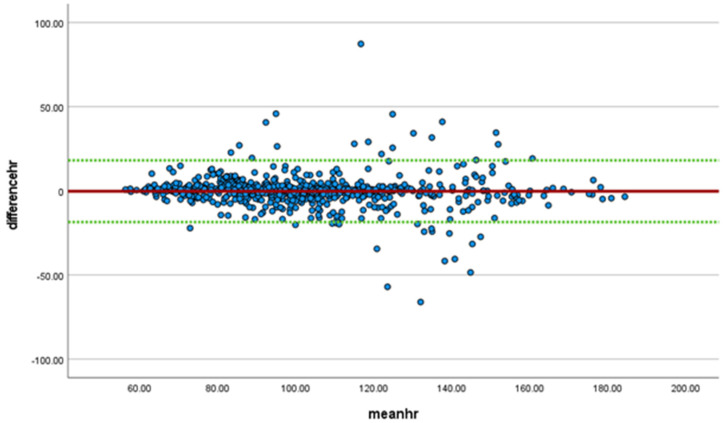
Bland-Altman Plot for heart rate comparing SS to Polar 10. Mean noted in red, 95% confidence interval in green.

**Figure 3 sensors-23-00877-f003:**
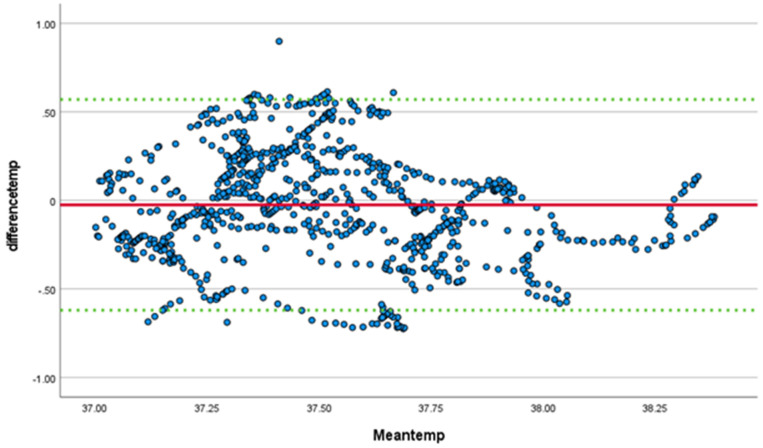
Bland-Altman test for temperature comparing SS to ingestible pill. Mean noted in red, 95% confidence interval in green.

**Table 1 sensors-23-00877-t001:** Participant characteristics.

Part ID	Temp	Sex	Age	Weight in	Weight out
40	mod	F	31	378.4	379.2
29	mod	F	21	159	160.2
52	mod	F	22	173	173
58	mod	F	23	146.2	146.6
89	mod	M	27	197.8	197.6
43	Mod	M	46	203.6	206.6
59	mod	F	39	178.4	179.8
61	mod	F	34	160.6	162.2
91	mod	M	28	206	206.4
62	mod	F	56	136	136
41	hot	F	22	136.8	136.6
93	hot	F	37	175.8	175
87	hot	F	38	198.4	198
64	hot	F	37	143.8	144.2
92	hot	F	23	168.8	169
84	hot	F	38	127.4	127.8
42	hot	F	42	139.2	139.4
75	hot	F	22	126	125.6
13	hot	F	22	188.8	189
28	hot	F	21	153.6	153.8
Average			31.45	174.88	175.3

**Table 2 sensors-23-00877-t002:** Collected data for all devices.

User	Peak Pill	Avg Pill	Peak SS	Avg SS	HR Max SS	HR Avg SS	HR Max P	HR Avg P
40			38.03	37.71	156	111.6	-	-
29	37.28	37.265	37.59	37.41	165	93.48	-	-
52	37.73	37.49	37.41	37.05	133	77	135.5	81.1
58	37.97	37.92	37.62	37.17	142	84.25	138	85.2
89	37.28	37.15	37.37	37.15	116	82.8	113.8	81.5
43	37.59	37.45	37.63	37.11	138	102.4	135.6	101.2
59	37.9	37.75	37.41	37.22	111	86.5	111.8	87.3
61	-	-	37.21	37.01	102	75.6	101	75.1
91	37.05	36.98	37.32	37.23	105	84	90	87
62	37.35	37.13	38.05	37.67	158	113.3	156	113.3
28	37.87	37.51	38.34	37.78	177	116.4	174.8	116
41	37.74	37.5	37.94	37.48	169	110	165.6	109
93	38.41	37.96	38.4	38.02	169	124	168.6	124.1
87	37.32	37.15	37.55	37.42	112	95.1	-	-
64	37.58	37.43	37.44	37.21	122	84.73	-	-
92	37.97	37.7	37.93	37.6	132	109.6	131.8	110
84	37.79	37.35	37.66	37.38	119	100.4	115.9	99.5
42	37.76	37.62	38.01	37.52	159	106.8	154.9	107.5
75	37.72	37.55	38.04	37.73	146	111.9	141.3	110
13	38.34	37.83	38.43	37.83	186	126.3	182.5	126.6
Overall	37.70	37.49	37.75	37.45	140.85	99.81	138.57	100.9

**Table 3 sensors-23-00877-t003:** Correlation and percentage difference of heart rate.

HR Participant	Pearson Correlation	SIG	ci			Percent Different Peak	Percent Different Average
40	No available data						
29	No available data						
52	0.945	<0.001	0.907–0.967			1.84	5.06
58	No available data						
89	0.953	<0.001	0.915–0.973			−1.93	−1.60
43	0.967	<0.001	0.928–0.984			−1.77	−1.19
59	0.788	<0.001	0.636–0.877			0.72	0.91
61	0.942	<0.001	0.894–0.967			−0.99	−0.67
91	0.332	0.012	0.073–0.545			−16.7	3.44
62	0.992	<0.001	0.986–0.995			−1.28	0
28	0.991	<0.001	0.985–0.994			−1.26	−0.34
41	0.698	<0.001	0.534–0.807			−2.05	−0.92
93	0.36	0.008	0.098–0.570			−0.24	0.08
87	No available data						
64	No available data						
92	0.974	<0.001	0.949–0.986			−0.15	0.36
84	0.966	<0.001	0.935–0.981			−2.67	−0.90
42	0.915	<0.001	0.857–0.949			−2.65	0.65
75	0.907	<0.001	0.833–0.947			−3.33	−1.73
13	0.982	<0.001	0.968–0.989			−1.92	0.24
overall	0.926	<0.001	0.915–0.935	0.961 (*p* < 0.001)	0.955–0.966	−1.65	1.08

**Table 4 sensors-23-00877-t004:** Correlation between Slate Safety and ingestible pill and percentage difference.

Temperature Participant	Pearson Correlation	SIG	ci	ICC		Percent Difference Peak	Percent Difference Average
40	No data available						
29	Two Data points only					−0.83	−0.39
52	Limited Data					0.85	1.17
58	1.00	<0.001	2 POINTS ONLY			0.92	1.98
89	0.821	<0.001	0.702–0.892			−0.24	0
43	0.497	<0.001	0.321–0.637			−0.11	0.91
59	0.922	<0.001	0.874–0.951			1.29	1.4
61	No data available						
91	0.747	<0.001	0.610–0.837			−0.73	−0.68
62	0.945	<0.001	0.912–0.965			−1.87	−1.45
28	0.96	<0.001	0.936–0.974			−1.24	−0.72
41	0.6	<0.001	0.447–0.717			−0.53	0.05
93	0.835	<0.001	0.748–0.891			0.03	−0.16
87	0.956	<0.001	0.854–0.985			−0.62	−0.73
64	0.186	0.725	−0.745–0.862 Limited data			0.37	0.59
92	0.961	<0.001	0.932–0.977			0.11	0.27
84	0.126	0.295	−0.111–0.348			0.34	−0.08
42	0.115	0.351	−0.128–0.343			−0.66	0.27
75	0.486	<0.001	0.239–0.669			−0.85	−0.48
13	0.903	<0.001	0.846–0.938			−0.23	0
overall	0.594	<0.001	0.550–0.635	0.742 (*p* < 0.001)	0.705–0.774	−0.13	0.09

## Data Availability

The data presented in this study are available on request from the corresponding author. The data are not publicly available due to IRB restrictions.

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
