# Peer review of "Comparison of Slate Safety Wearable Device to Ingestible Pill and Wearable Heart Rate Monitor"

_sensors, 2023, doi:10.3390/s23020877_

Round 1
Reviewer 1 Report
The manuscript describes a randomized controlled study, comparing the diagnostic accuracy of a wearable multiparameter monitor to other wearable or ingestible single-parameter monitors. The study design was adequate for the stated goal. More details will need to be added to clarify the significance of the findings.
1. Describe the technical aspects and setup of the SS device in more detail. An image or photo of the device in use would be helpful. To increase the relevance of the study and assess the device's utility in real world applications, describe parameters such as maximum duration of monitoring, sensor technology, sampling rate and range of measurements. For remote monitoring, describe factors such as the setup, time interval of transmissions, possibility of automated alerts.
2. Data for the 3.1 comfort section should be presented in more detail in the results section as this was one of the goals of the study.
3. How does this device compare to other wearable multiparameter, remotely monitored physiologic sensors? The introduction and discussion sections will need to be expanded to present data from similar studies, evaluating the utility and limitations of this technology. What does the SS device offer, compared to similar devices?
Author Response
|
The manuscript describes a randomized controlled study, comparing the diagnostic accuracy of a wearable multiparameter monitor to other wearable or ingestible single-parameter monitors. The study design was adequate for the stated goal. More details will need to be added to clarify the significance of the findings. |
|
1. Describe the technical aspects and setup of the SS device in more detail. An image or photo of the device in use would be helpful. To increase the relevance of the study and assess the device's utility in real world applications, describe parameters such as maximum duration of monitoring, sensor technology, sampling rate and range of measurements. For remote monitoring, describe factors such as the setup, time interval of transmissions, possibility of automated alerts.
Additional information and picture added Lines 141-151
|
|
2. Data for the 3.1 comfort section should be presented in more detail in the results section as this was one of the goals of the study.
Further detail was added to under study materials and methods section (line 168-171), the results section (line 187-197) |
|
3. How does this device compare to other wearable multiparameter, remotely monitored physiologic sensors? The introduction and discussion sections will need to be expanded to present data from similar studies, evaluating the utility and limitations of this technology. What does the SS device offer, compared to similar devices? |
|
Additional information has been added to the introduction (line 53-77) and in the discussion (line 269-282, and 306-323 and 346-351) |
Reviewer 2 Report
In the submitted manuscript, Callihan et al. demonstrates the use of combination of multiple wearable devices to monitor heat stress in means of heart rate and body temperature. The article is technically sound and would be ready to publish after the justification of the below concerns:
1) For the statistical accuracy, authors need to justify the results with respec to the fact that the majority (85%) of the participants are female. In some of the office-based studies it was found that women tend to feel the cold more than their male co-workers. Authors need to discuss in more detailed way about the gender difference and possible effect of that considering the heat susceptibility could be different.
2) ‘‘According to Polar's testing against medical-grade heart rate equipment and other heart rate sensors - the ECG H10 sensor detects HR within 2 ms accuracy at 92.9% for running, 99.3% for cycling, 95.3% for weight training, 95.6% for all activities combined.’’ Therefore the activity of each individual seems to affect the result differently. Authors should justify the possible accuracy related alterations depending on each individual activity and what nursing activity actually includes.
3) Since the HR values varies considerably among the participants, authors are suggested to use Bland-Altman plot analysis for Table 3, where the difference of each method is plotted against the mean of each measurement. Although the correlation coefficient is significant, linear correlations for the considerably varying parameters do not justify the methods could be used interchangeably. Same applies for temperature measurements in Table 4.
Author Response
|
In the submitted manuscript, Callihan et al. demonstrates the use of combination of multiple wearable devices to monitor heat stress in means of heart rate and body temperature. The article is technically sound and would be ready to publish after the justification of the below concerns: |
|
1) For the statistical accuracy, authors need to justify the results with respec to the fact that the majority (85%) of the participants are female. In some of the office-based studies it was found that women tend to feel the cold more than their male co-workers. Authors need to discuss in more detailed way about the gender difference and possible effect of that considering the heat susceptibility could be different.
We agree that there is a difference in the physiological response to temperature based on sex. This will need to be addressed with any prevention program developed when using any system to ensure adequate protection against heat stress. This study looked psecifically at correlating the SS to the ingestible pill system. While the differences in the physiological response can be seen at a low level based on the sample size, sex did not have significance in the the correlations between the systems. Additional information was added in the discussion section to address this. Line 257-263 |
|
2) ‘‘According to Polar's testing against medical-grade heart rate equipment and other heart rate sensors - the ECG H10 sensor detects HR within 2 ms accuracy at 92.9% for running, 99.3% for cycling, 95.3% for weight training, 95.6% for all activities combined.’’ Therefore the activity of each individual seems to affect the result differently. Authors should justify the possible accuracy related alterations depending on each individual activity and what nursing activity actually includes.
Added a section under Data sources/ measurements to adress this. Also added description of nursing activites to study design section Line 135-140 |
|
3) Since the HR values varies considerably among the participants, authors are suggested to use Bland-Altman plot analysis for Table 3, where the difference of each method is plotted against the mean of each measurement. Although the correlation coefficient is significant, linear correlations for the considerably varying parameters do not justify the methods could be used interchangeably. Same applies for temperature measurements in Table 4.
Bland-Altman plots added for both temperature and heart rate. There is a demonstrated bias toward the SS reading higher for both temperature and heart rate, which is preffered to a bias in the other direction. Lines 211-215, 235-239, and 286-289 |
Round 2
Reviewer 1 Report
The authors have added the requested clarifications to the manuscript.
The updated reference list will need to be corrected as several references are incomplete.